# Synthesis, DFT Calculations, Antiproliferative, Bactericidal Activity and Molecular Docking of Novel Mixed-Ligand Salen/8-Hydroxyquinoline Metal Complexes

**DOI:** 10.3390/molecules26164725

**Published:** 2021-08-04

**Authors:** Badriah Saad Al-Farhan, Maram T. Basha, Laila H. Abdel Rahman, Ahmed M. M. El-Saghier, Doaa Abou El-Ezz, Adel A. Marzouk, Mohamed R. Shehata, Ehab M. Abdalla

**Affiliations:** 1Chemistry Department, Faculty of Girls for Science, King Khalid University, Abha 61421, Saudi Arabia; 2Chemistry Department, College of Science, University of Jeddah, Jeddah 21959, Saudi Arabia; marmarek-3@hotmail.com; 3Chemistry Department, Faculty of Science, Sohag University, Sohag 82534, Egypt; el.saghier@science.sohag.edu.eg; 4Pharmacology and Toxicology Department, Faculty of Pharmacy, October University for Modern Sciences and Arts (MSA University), Giza 12556, Egypt; Dabulez@msa.edu.eg; 5Department of Pharmaceutical Chemistry, Faculty of Pharmacy, Al-Azhar University, Assiut Branch, Assiut 71524, Egypt; adel_marzouk77@yahoo.com; 6Chemistry Department, Faculty of Science, Cairo University, Giza 12613, Egypt; mrshehata_05@hotmail.com; 7Chemistry Department, Faculty of Science, New Valley University, Alkharga 72511, Egypt; ehababdalla99@yahoo.com or

**Keywords:** anticancer, antimicrobial, mixed-ligand, salen, 8-hydroxyquinoline, quantum calculation

## Abstract

Despite the common use of salens and hydroxyquinolines as therapeutic and bioactive agents, their metal complexes are still under development. Here, we report the synthesis of novel mixed-ligand metal complexes (MSQ) comprising salen (S), derived from (2,2′-{1,2-ethanediylbis[nitrilo(E) methylylidene]}diphenol, and 8-hydroxyquinoline (Q) with Co(II), Ni(II), Cd(II), Al(III), and La(III). The structures and properties of these MSQ metal complexes were investigated using molar conductivity, melting point, FTIR, ^1^H NMR, ^13^C NMR, UV–VIS, mass spectra, and thermal analysis. Quantum calculation, analytical, and experimental measurements seem to suggest the proposed structure of the compounds and its uncommon monobasic tridentate binding mode of salen via phenolic oxygen, azomethine group, and the NH group. The general molecular formula of MSQ metal complexes is [M(S)(Q)(H_2_O)] for M (II) = Co, Ni, and Cd or [M(S)(Q)(Cl)] and [M(S)(Q)(H_2_O)]Cl for M(III) = La and Al, respectively. Importantly, all prepared metal complexes were evaluated for their antimicrobial and anticancer activities. The metal complexes exhibited high cytotoxic potency against human breast cancer (MDA-MB231) and liver cancer (Hep-G2) cell lines. Among all MSQ metal complexes, CoSQ and LaSQ produced IC_50_ values (1.49 and 1.95 µM, respectively) that were comparable to that of cisplatin (1.55 µM) against Hep-G2 cells, whereas CdSQ and LaSQ had best potency against MDA-MB231 with IC_50_ values of 1.95 and 1.43 µM, respectively. Furthermore, the metal complexes exhibited significant antimicrobial activities against a wide spectrum of both Gram-positive and -negative bacterial and fungal strains. The antibacterial and antifungal efficacies for the MSQ metal complexes, the free S and Q ligands, and the standard drugs gentamycin and ketoconazole decreased in the order AlSQ > LaSQ > CdSQ > gentamycin > NiSQ > CoSQ > Q > S for antibacterial activity, and for antifungal activity followed the trend of LaSQ > AlSQ > CdSQ > ketoconazole > NiSQ > CoSQ > Q > S. Molecular docking studies were performed to investigate the binding of the synthesized compounds with breast cancer oxidoreductase (PDB ID: 3HB5). According to the data obtained, the most probable coordination geometry is octahedral for all the metal complexes. The molecular and electronic structures of the metal complexes were optimized theoretically, and their quantum chemical parameters were calculated. PXRD results for the Cd(II) and La(III) metal complexes indicated that they were crystalline in nature.

## 1. Introduction

Metal complexes constitute an important class of materials and are encountered daily. Metal complexes show remarkable stability compared to their ligands and metals alone, owing to the chelate effect [1]. An informed selection of metal conductors and ligand bridges allows for the application of such metal complexes in adsorption, ion exchange, and separation [2]; biomimetic and heterogeneous catalysis [3,4]; biomedical applications; sensor technologies [5]; luminescence [6,7,8]; proton conductivity [9]; and drug delivery [7,8]. The structural study of mixed-ligand metal complexes can provide information on how biological systems attain their stability and specificity [10]. Accordingly, we have selected two metal complexing agents, namely, salen and 8-hydroxyquinoline ligands, to prepare new mixed-ligand metal complexes.

Salen (S) is an excellent ligand that contains N_2_O_2_ donor sites through which it readily binds to metal ions forming metal complexes with octahedral, tetrahedral, or square planar geometries [11,12,13]. The large numbers of studies on salen highlight its importance [14,15,16,17,18,19,20,21,22,23,24]. 8-Hydroxyquinoline (typically abbreviated to Q in its metal complexes) and its derivatives are found in plants and have also been prepared synthetically. Owing to their chelating ability towards many metal cations, hydroxyquinoline derivatives have found many applications. For instance, they are used as preservatives in the textile industry and as fungicides in the agricultural, paper, and wood industries [25]. Furthermore, they are widely used in metal–metal complexation research and as antibacterial, insecticidal, neuroprotective, and anti-HIV agents in medicine [26,27].

Motivated by the uncommon tridentate salen binding in its mixed-ligand metal complexes with imidazole MSI, together with our growing interest in the properties/applications of Schiff base metal complexes, we report in this contribution the synthesis, photophysical properties, and DFT calculations of new mixed-ligand of salen/8-hydroxyquinoline metal complexes (MSQ; M = Co(II), Ni(II), and Cd(II), or Al(III) and La(III) ions). Analytical and theoretical (DFT) structural characterizations revealed a mono basic tridentate coordination mode of the salen ligand and bidentate for 8-hydroxyquinoline to afford octahedral geometries of the general formula [M(S)(Q)(X)], X = H_2_O or Cl. Importantly, the antimicrobial activities of the prepared MSQ metal complexes were evaluated and compared with those of their salen/imidazole MSI analogues. Additionally, the cytotoxicity of the MSQ metal complexes against the liver carcinoma (Hep-G2) and the breast cancer (MDA-MB231) carcinoma cell lines were investigated.

## 2. Materials and Methods

### 2.1. Reagents

Ethylene diamine, salicylaldehyde, 8-hydroxyquinoline, sodium hydroxide, lanthanum(III) chloride (LaCl_3_7H_2_O), aluminum(III) chloride (AlCl_3_6H_2_O), cadmium(II) chloride (CdCl_2_H_2_O), nickel(II) chloride (NiCl_2_6H_2_O), and cobalt(II) chloride (CoCl_2_6H_2_O) were purchased from (Sigma Aldrich Chemie GmbH, München, Germany). Organic solvents, including absolute ethanol and dimethyl sulfoxide (DMSO), were provided at reagent grade, and used without purification.

### 2.2. Characterization

The melting and decomposition points of salen and its mixed 8-hydroxyquinoline metal complexes were measured on a Gallen Kamp apparatus (Nikon Corporation, Tokyo, Japan). FTIR spectra of the metal complexes in the region 400 to 4000 cm^−1^ were recorded from KBr pellets using a Shimadzu FTIR spectrophotometer model 8101 (Shimadzu Corporation, Kyoto, Japan). A JENWAY model 4320 conductivity meter (Pontypool, UK) was used to measure the molar conductance of the metal complexes. UV–VIS spectra were recorded in DMSO using a Jasco model V-530 spectrophotometer (JASCO, Easton, MD, USA). FT-NMR spectrometer (Bruker ARX 400.1) (Billerica, MA, USA) at 400 MHz (^1^H) and 100.6 MHz (^13^C) was used to record ^1^H NMR and ^13^C NMR spectra in DMSO-*d*_6_ utilizing TMS as an internal standard (Sohag University, Faculty of Science, Chemistry Department Central Lab.). Mass spectra were obtained with an (MS-5988 GS-MS, USA) Hewlett-Packard instrument at the Microanalytical Center, National Center for Research, Dokki, Egypt, using electrospray ionization at 70 eV. Elemental analyses of salen and its metal complexes were performed in Cairo University with a Perkin-Elmer 240c elemental analyzer. Magnetic measurements were performed using a Guy’s balance. Additionally, Shimadzu corporation 60 H analyzer was utilized to obtain the thermograms of the metal complexes in air with a heating rate of 10 °C/min from ambient temperature to 800 °C. A HANNA 211 pH meter (Merck KGaA, Darmstadt, Germany) was used to measure the pH values using Britton universal buffers. Anticancer activity was evaluated using the ELISA technique, and a microplate reader (Meter tech. Σ 960, Winooski, VT, USA) was used to measure the absorbance results at 564 nm.

### 2.3. Synthesis of Salen

In a round flask, 0.601 g (0.01 mol) ethylenediamine and 2.44 g (0.02 mol) salicylaldehyde were mixed in 50 mL absolute ethanol and refluxed for 60 min. The reaction mixture was concentrated to half of its volume, whereupon the yellow precipitate formed was isolated by filtration, washed with ethanol, and dried at laboratory temperature [28]. Yield 96%; M. p. 127 °C; FTIR (KBr, cm^−1^): 3292 (OH), 3049–3007 (CH)_arom_, 2899–2867 (CH)_aliph_, 1608 (C=N), 1217 (C-C); ^1^H NMR (DMSO-*d*_6_/D_2_O, 400 MHz): 3.87 (t, 4H, N-CH_2_-CH_2_-N), 6.69–7.37 (m, 8H, Ar–H), 8.54 (s, 2H, 2 N=CH), 13.3 (br s, 2H, 2 ArOH); ^13^C NMR (100 MHz, DMSO-*d*_6_): 58.68, 116.99, 119.00, 132.22, 132.94, 161.19, 167.28.

### 2.4. Synthesis of Mixed Metal Complexes (MSQ)

Into a boiling flask, we added 1 mmol of the metal salt (0.238 g of Co(II), 0.236 g of Ni(II), 0.201 g of Cd(II), 0.241 g of Al(III), or 0.372 g of La(III)) and 0.268 g (1 mmol) of salen in 10 mL of an aqueous ethanolic mixture. The reaction mixture was then refluxed at 79 °C with constant stirring, and after 1 h, 0.145 g (1 mmol) of 8-hydroxyquinoline in 10 mL ethanol was added dropwise. The reaction was then refluxed for another 1 h. The resulting reaction mixture was left to cool and evaporate overnight. The colored solid products (dark brown for CoSQ, brownish-red for NiSQ, dark-yellow for CdSQ, light-yellow for AlSQ, and dark-yellow for LaSQ) were filtered from the reaction mixture, thoroughly washed with ethanol to remove any traces of unreacted starting materials, and dried under vacuum. The purity of the metal complexes was checked by TLC [29,30] (Scheme 1). The yields, melting points, conductance values, magnetic moments, and elemental analysis results for the mixed-ligand metal complexes are shown in (Table 1).

### 2.5. Computational Study

The lowest energy geometries of the ligands and their metal complexes were derived using DFT at the DFT/B3LYP/6-311++g(d,p) level of theory for C, H, N, and O atoms and LANL2DZ for metals, utilizing the Gaussian09 program [31]. IR Frequency calculations were performed on the optimized geometries, showing that all the converged states correspond to true minima (no imaginary frequencies). 

### 2.6. Molecular Docking Study

Molecular docking studies were performed using MOA2014 software [32] in order to explore the possible binding modes for the most active site of the receptor of breast cancer oxidoreductase (PDB ID: 3HB5). The crystal structures of the 3HB5 breast cancer receptor were downloaded from the Protein Data Bank (http://www.rcsb.org./pdb accessed date 21 April 2021). The protein was prepared for the docking as follows: (i) The ligand molecule was removed from the enzyme active site. (ii) Hydrogen atoms were added to the proteins with MOE and minimized, keeping all the heavy atoms fixed until RMS gradient of 0.01 kcal/mol and RMS distance of 0.1 Å were reached. (iii) Partial charges were computed using MMFF94x force field. (iv)The structures of salen and its metal complexes were created in PDB file format using the Gaussian09 software package. (v) The structures were subjected to energy minimization using MMFF94x force field, and the partial charges were computed using the same force field. (vi) Docking calculations were performed using Alpha triangle placement method, and poses were ranked by London dG scoring method.

### 2.7. PXRD Analysis

PXRD analysis of the metal complexes was performed using a known standard method [33].

The average crystallite size (ξ) was calculated from the PXRD pattern according to the Debye–Scherrer equation [34,35]
$$\xi =\frac{K\lambda }{\beta 1/2\;\cos\theta }$$
where λ is the wavelength of the X-rays (1.542475 Å), K is a constant taken as 0.95 for organic compounds [34], and β1/2 is the width at half maximum of the reference diffraction peak measured in radians.

### 2.8. Antimicrobial Potency

The biological activities of the metal complexes were evaluated using two Gram-positive bacteria (*Bacillus subtilis* (+ve) and *Staphylococcus aureus* (+ve)), two Gram-negative bacteria (*Escherichia coli* (−ve) and *Proteus vulgaris* (−ve)), and two fungi (*Candida albicans* and *Aspergillus flavus*) by the agar diffusion test. In this method, we measured the effectiveness of antibiotics on a specific microorganism. An agar plate was first spread with bacteria, and then paper disks of antibiotics were placed atop of it. The bacteria were then allowed to grow on the agar media and then observed for growth and effect of the antibiotic on it. The amount of space around every antibiotic disk indicated the lethality of that antibiotic on the bacteria in question. Highly active antibiotics will show a large zone of no bacterial development, while an ineffectual antibiotic will display bacterial growth around the disc. The compound under investigation was dissolved in DMSO [36], and paper discs were impregnated with the solution. The disks were dried and set in agar plates containing the microorganisms. Then, the plates were incubated for 25–33 h at 24 ± 2 °C, and the inhibition zones, i.e., those where the concentration of the compound exceeds the minimum inhibitory concentration (MIC), were accurately evaluated. Gentamicin as an antibacterial agent and ketoconazole as an antifungal agent were used for comparison.

### 2.9. MTT Assay

The cytotoxic activities of the synthesized ligands and metal complexes against the breast cancer cell line MDA-MB231, the hepatic cellular (Hep-G2), and normal cell line (HEK-293) were assessed by MTT assay. A cell suspension was diluted by complete medium to a concentration of 5 × 10^4^ cell/mL. Using a micropipette, we pipetted 100 μL aliquots of the cell suspension into each well of 96-well plate (≈5000 cells/well). The 96-well plate was incubated at 37 °C for 24 h to allow cell attachment. After 24 h, cells were treated with 100 µL of growth medium containing 0, 0.001, 0.01, 0.1, 1, 10, or 100 μL of the newly synthesized compounds in triplicate. Cells were washed with phosphate-buffered saline (PBS), and fresh relevant medium containing 20 μL MTT in PBS (0.5 mg/mL) was added to the test wells. The plate was further incubated in a CO_2_ incubator at 37 °C for 4 h, and then MTT assay analysis was performed. The MTT assay technique is based on the reduction of the tetrazolium salt MTT to insoluble purple formazan by metabolically active cells, making their activities quantifiable by spectrophotometry. Accordingly, the formed formazan crystals were dissolved in 120 μL DMSO for each well. Cell viability was determined by measuring the absorbance of each well at 570 nm (and at a reference wavelength of 630 nm) using an ELISA plate reader. Results are expressed in terms of the concentration required to inhibit cell growth by 50% relative to untreated cells (IC_50_). IC_50_ values were calculated using Graph Pad Prism version 6.01, 2012 (GraphPad software, San Diego, CA, USA), by plotting the log concentration versus corresponding viability (%) to generate dose–response curves.

## 3. Results and Discussion

### 3.1. Physicochemical Properties

All the metal chelates were colored and stable towards air and moisture. The analytical results for the metal complexes were consistent with their proposed molecular formula and confirm the formation of 1:1:1 mixed-ligand salen/8-hydroxyquinoline metal complexes. The values of molar conductance for the metal complexes in 10^−3^ M DMF solutions fell in the range 24.10–53.30 Ω^−1^ cm^2^ mol^−1^, except that of AlSQ (103.50). These results demonstrate that the solutions are non-electrolytic except, that of the Al metal complex, which is ionic.

### 3.2. ^1^H NMR and ^13^C NMR Spectra

The NMR of salen [37] and 8-hydroxyquinoline [38,39] were previously studied. Comparing the positions of the proton signals for mixed salen/8-hydroxyquinoline with those for the Cd(II) and Al(III) metal complexes shown in (Figure 1A–C and Appendix A), we were able to conclude that all the signals occurred in their expected positions and were shifted only slightly upon the coordination of the salen ligand to the metal ions [40]. The 4H multiplet signals around 3.87 ppm may be assigned to the ethylene group (CH_2_CH_2_) of the salen ligand. These signals appeared at 2.51–2.81 and 3.83–4.05 ppm in the [Cd(S)(Q)(H_2_O)] and [Al(S)(Q)(H_2_O)]^+^Cl^−^ metal complexes, respectively.

The multiplet signals in the region 6.82–7.54 ppm for the salen ligand may be assigned to aromatic protons [41]. These signals were found at 6.90–7.47 and 6.66–7.36 ppm for [Cd(S)(Q)(H_2_O)] and [Al(S)(Q)(H_2_O)]^+^Cl^−^. These signals may also be assigned to quinoline. Other signals appeared at 7.49–8.71 and 7.49–8.62 ppm with an integral value of 6H (4H cyclic, 2H olefin) for [Cd(S)(Q)(H_2_O)] and [Al(S)(Q)(H_2_O)]^+^Cl^−^, respectively, and 1H NH for complex [Al(S)(Q)(H_2_O)]^+^Cl appeared at 13.34 ppm.

The 2H singlet signal observed at 8.54 ppm for the salen ligand may be assigned to the two N=CH protons. This signal slightly shifted to 8.62 and 8.71 ppm for the [Cd(S)(Q)(H_2_O)] and [Al(S)(Q)(H_2_O)]^+^Cl^−^ metal complexes, respectively. This may be attributed to the coordination of different metal ions through the azomethine group. The singlet signal observed at 13.3 ppm in the ^1^H NMR spectrum of salen, which corresponded to the phenolic proton, was not observed for the [Cd(S)(Q)(H_2_O)] and [Al(S)(Q)(H_2_O)]^+^Cl^−^ metal complexes, while new signals for NH in the metal complexes were observed at 7.49–8.71 and 7.49–8.62 ppm, respectively (1H for NH Cd and Al).

From the ^1^H NMR results, we found that the metal complexes contained some water molecules, wherein the 2H signal observed at 3.17 and 3.30 ppm could be assigned to the OH of H_2_O. It may also be concluded that quantum calculation, analytical, and experimental measurements seemed to suggest the proposed structure of the compounds and the different metal ions coordinated with salen through the azomethine nitrogen and phenolic oxygen from one side, and the carbonyl and NH group from the other side. As the NH group coordinated with the metal ion, salen behaved as a univalent tridentate ligand (Table 2).

Moreover, the ^13^C NMR spectrum of the salen ligand (Appendix A) showed signals for two CH_2_ groups at 58.96, aromatic carbons at 116.99 to 161.19, and the azomethine group CH=N at 167.28. Upon metal complexation with Cd and Al to form CdSQ and AlSQ, the obtained 13C NMR spectrum (Appendix A) revealed the disappearance of the azomethine group signal and the appearance of signals for cyclic C=O at 158.98 and 167.37, respectively.

When we calculated the energy of 5-coordinated Cd geometry, the metal complex produced higher energy −1403.037 Hartree, compared to −1479.550 Hartree for the octahedral one. Moreover, 5-coordinated Al metal complex produced higher energy −1357.168 Hartree, compared to −1433.425 Hartree for the octahedral one. Similar results wer found for other metal complexes (Appendix A).

### 3.3. IR Spectra

The FTIR spectra of salen and its metal complexes are shown in (Figure 2 and Appendix A). The spectrum for salen featured a band at 1608 cm^−1^, which corresponded to the −C=N stretching vibration. Upon the formation of mixed salen/8-hydroxyquinoline metal complexes of Co(II), Ni(II), Cd(II), Al(III), and La(III), this band was shifted to a higher frequency (1637, 1626, 1633, 1634, and 1621 cm^−1^, respectively). This degree of red shift was evidence of the participation of the azomethine nitrogen atoms in metal complex formation [33,42,43]. The salen ligand also presented a band at 1247 cm^−1^, which was assigned to the ν(C–O) stretching vibration. The coordination of the phenolic oxygen atom was also confirmed by the red shift (1231–1207 cm^−1^) of the ν(C–O) peak upon metal complex formation. This was supported by the appearance of bands at 445, 441, 494, 485, and 484 cm^−1^ and bands at 527, 538, 578, 545, and 533 cm^−1^ corresponding to the stretching vibrations of the M–N and M–O bonds for the Co(II), Ni(II), Cd(II), Al(III), and La(III) metal complexes, respectively.

In the FTIR spectra of the mixed salen/8-hydroxyquinoline metal complexes, the appearance of a band at 1740 cm^−1^ for all the metal complexes was attributed to the C=O stretching vibration, while the bands at 3173, 3229, 3137, 3191, and 3142 cm^−1^, respectively, were due to the NH group. The IR spectra of all the prepared metal chelates showed broad bands at 3432 and 3449 cm^−1^, which could be assigned to the ν(OH) stretching vibration of hydrated water molecules in the mixed-ligand Ni(II) and La(III) metal complexes. The IR spectra of the Co(II), Ni(II), Cd(II), and Al(III) metal complexes showed bands at 950, 903, 976, and 980 cm^−1^, respectively, which were assigned to the rocking mode of coordinated water (Table 3).

### 3.4. Electronic Spectra

The UV–VIS spectra of the salen ligand and its metal complexes in DMSO were measured at room temperature in the region 200–700 nm and are shown in Figure 3. The absorption spectrum of the salen ligand featured three absorption bands at 280, 320, and 409 nm. The first high-intensity band appeared at λ_max_ = 280 nm may be attributed to the π→π* transition of the aromatic rings. The second and third absorption bands that appeared at λ_max_ = 320 and 409 nm can be attributed to the n→π* transition of the azomethine group (C=N) and charge transfer, respectively [44,45].

Compared to those of the free ligand, the electronic spectra of the metal complexes showed bands that were shifted to 274–291 nm and 304–363 nm for the π→π* and n→π* transitions, respectively, confirming the coordination of the azomethine nitrogen to the metal ions. The appearance of a 405 nm in the Cd(II) metal complex and at 415 nm for La(III) metal complex corresponded to charge transfer from ligand to metal LMCT. Furthermore, absorption bands in the visible region at 408 nm for the Co(II) and 415 nm for the Ni(II) metal complexes were observed. These bands were considered to arise from d–d transitions [46].

### 3.5. pH Profiles

The pH profiles (i.e., absorbance vs. pH) shown in Figure 4 exhibited typical dissociation curves and revealed high stabilities in the pH range 5–10 for the chelates of Co(II) and Cd(II), 6–10 for Ni(II), 5–11 for Al(III), and 6–11 for La(III). This revealed that the formation of the metal chelate greatly stabilized the ligands. Consequently, a suitable pH range for the application of the resultant mixed-ligand metal chelates was found to be 5–11.

### 3.6. ESI-MS Spectra

MS has become increasingly used for elucidation of the molecular structures of ligands and their metal complexes. The mass spectra of salen and its mixed 8-hydroxyquinoline metal complexes of Co(II), Ni(II), Cd(II), Al(III), and La(III) showed molecular ion peaks at *m*/*z* 269.07 for salen and at *m*/*z* 488.40, 497.16, 541.88, 491.95, and 603.83 for its metal complexes, respectively. These data are in good agreement with the proposed molecular formulae. The ESI-MS spectra of the metal complexes are shown in Figure 5 and Appendix A. The suggested fragmentation pattern for CdSQ is shown in Appendix A.

### 3.7. PXRD Analysis

The growth of single crystals of the synthesized compounds failed, and hence PXRD was performed. The powder diffraction patterns of salen and metal complexes of Cd(II) and La(III) with salen and mixed salen/8-hydroxyquinoline were recorded over the 2θ range 5°–70° range (Table 4). The position of the highest intensity peak was determined, along with the width of this peak at half maximum and the d-spacing. The diffractograms of the ligand and metal complexes are shown in Figure 6 and Appendix A. The diffractogram of the salen ligand featured a reflection with its maxima at 2θ = 13.14° corresponding to a d-spacing value of 6.7318.

The PXRD patterns of the metal complexes were completely different from that of salen, demonstrating the formation of the coordination compounds. The diffraction pattern revealed well-defined crystalline peaks, indicating the crystalline nature of salen and the CdSQ and LaSQ metal complexes. The average particle size of the crystalline metal complexes was calculated using Scherrer’s formula. The average particle sizes for salen ligand and its Cd(II) and La(III) mixed 8-hydroxyquinoline metal complexes were calculated to be 0.521, 0.423, and 0.343 nm, respectively.

### 3.8. Thermogravimetric Analysis

The stepwise decompositions of the resultant metal complexes with respect to temperature and the formation of the respective metal are depicted in Figure 7 and Appendix A. The thermograms of the metal chelates indicated the presence of one coordinated water molecule in all the metal complexes except LaSQ, which contained one hydrated water molecule, and NiSQ, which contained a half equivalent hydrated water molecule.

The thermal degradation of the hydrated metal chelates involves the loss of hydration water molecules followed by loss of coordinated water molecules and degradation of the ligand molecules in later stages, as displayed in Table 5 and Scheme 2 and Appendix A.

The thermograms of the CoSQ and LaSQ metal complexes showed six decomposition steps within the temperature range 35–740 °C. The first steps of degradation within the temperature range 35–250 °C corresponded to the loss of coordinated and hydrated water molecules with mass losses of 3.65% and 3.32% (calc. 3.68% and 2.98%), respectively. The second steps of decomposition occurred within the temperature range 200–290 °C with mass losses of 5.70% and 21.31% (calc. 5.73% and 21.28%) that indicated the removal of C_2_H_4_ and C_5_H_3_NOCl, respectively. The third stages occurred in the temperature range 265–395 °C with mass losses of 13.15% and 8.81% (calc. 13.10% and 8.61%), respectively. This corresponded to loss of C_4_H_2_N and C_4_H_4_, respectively. The fourth stages occurred in the temperature range 320–490 °C with mass losses of 31.90% and 7.40% (calc. 31.94% and 7.28%), respectively. This corresponded to loss of C_10_H_6_NO and N_2_O, respectively. The fifth stages occurred in the temperature range 360–580 °C with mass losses of 10.60% and 5.10% (calc. 10.64% and 4.97%), respectively. This corresponded to loss of C_4_H_4_ and CH_2_O, respectively. The sixth stages occurred in the temperature range 385–740 °C with mass losses of 22.74% and 30.96% (calc. 22.72% and 31.79%), respectively. This corresponded to loss of C_5_H_5_NO_2_ and C_15_H_12_, leaving Co and La as residues, respectively.

The thermograms of the NiSQ, and CdSQ metal complexes showed four decomposition steps in the temperature range 25–665 °C. The first steps of degradation from 25 to 240 °C corresponded to the loss of hydrated and coordinated water molecules with mass losses of 3.70%, and 3.30% (calc. 5.43%, and 3.32%), respectively. The second steps of decomposition within the temperature range 200–340 °C with mass losses of 41.85%, and 21.38% (calc. 41.80%, and 21.44%) indicate the removal of C_11_H_12_N_2_O_2_, and C_5_H_10_NO_2_, respectively. The third stages in the temperature range 290–465 °C with mass losses of 26.61%, and 6.10% (calc. 26.63%, and 5.54%) corresponded to loss of C_8_H_4_NO, and CH_2_O, respectively. The fourth stages in the temperature range 315–655 °C with mass losses of 15.65%, and 48.47% (calc. 15.77%, and 48.90%) corresponded to loss of C_6_H_5_, and C_19_H_9_N_2_, leaving Ni, and Cd as residues, respectively.

The thermogram of the AlSQ metal complex showed five decomposition steps within the temperature range 90–710 °C. The first step of degradation within the temperature range 90–200 °C corresponded to the loss of a coordinated water molecule, hydrogen chloride, and nitrogen dioxide with a mass loss of 19.9% (calc. 20.42%). The second step of decomposition within the temperature range 200–325 °C with a mass loss of 14.20% (calc. 14.34%) indicated the removal of C_3_H_5_NO. The third stage in the temperature range 325–400 °C with a mass loss of 25.85% (calc. 26.01%) corresponded to loss of C_9_H_6_N. The fourth stage in the temperature range 400–490 °C with a mass loss of 10.55% (calc. 10.57%) corresponded to loss of C_4_H_4_. The fifth stage in the temperature range 490–410 °C with a mass loss of 22.72% (calc. 22.69%) corresponded to loss of C_9_H_5_, The sixth stage in the temperature range 580–740 °C with a mass loss of 22.74% (calc. 22.72%) corresponded to loss of C_5_H_5_NO_2_, leaving Co as a residue.

### 3.9. DFT Analysis

#### 3.9.1. Molecular DFT Calculation of Salen

The ligand has two isomers, keto-enol form (A) and enol-enol form (B) (Figure 8). The keto-enol form is more stable than enol-enol form by −0.0073 Hartree, −0.1986 ev, −4.5808 kcal/mol, or −19.1661 kJ/mol. The hydrogen bond between C=O and N–H is shorter (stronger) in (A) than the hydrogen bond between O–H and N in (B).

Figure 8, shows the optimized structures of the ligand (A) as the lowest energy configurations. The natural charges obtained from natural bond orbital analysis (NBO) show that the more negative active sites are in the order of O2 (−0.728) > O1 (−0.720) > N1 (−0.488) > N2 (+0.583). Thus, the metal ions prefer tridentate coordination to O2, N1, and N2, forming stable 5- and 6-membered rings [37].

Figure 9 and Table 6 show the optimized structures and some selected bond distances and angles of the mixed-ligand metal complexes.

#### 3.9.2. [Co(S)(Q)(H_2_O)]

Figure 9A shows the optimized structures of the metal complex [Co(S)(Q)H_2_O)] as the lowest-energy configurations. The Co atom is six-coordinate in an octahedral geometry, with N1, N2, N3, and O4 being coplanar with a 6.999° deviation (Appendix A).

The N1**•••**N2, N1**•••**O2, and N2**•••**O2 distances in the free ligand (3.788, 4.261, and 7.003 Å) are longer than those in the metal complex (2.709, 2.720, and 3.045 Å) due to metal complex formation through N1, N2, and O2.

The natural charges on the coordinated atoms as calculated using natural bond orbital (NBO) analysis are Co (+0.409), O2 (−0.650), N1 (−0.487), N2 (−0.653), N4 (−0.498), N5 (−0.490), and Cl (−0.433).

#### 3.9.3. [Ni(S)(Q)(H_2_O)]

Figure 9B shows the optimized structures of the metal complex [Ni(S)(Q)(H_2_O)] as the lowest-energy configurations. The Ni atom is six-coordinate in an octahedral geometry, with N1, N2, N3, and O4 being coplanar with a −3.917° deviation (Appendix A).

The N1**•••**N2, N1**•••**O2, and N2**•••**O2 distances in the free ligand (3.788, 4.261, and 7.003 Å) are longer than those in the metal complex (2.777, 2.717, and 2.950 Å) due to metal complex formation through N1, N2, and O2.

The natural charges on the coordinated atoms derived by NBO analysis are Ni (+0.779), O2 (−0.655), N1 (−0.541), N2 (−0.700), N3 (−0.528), O4 (−0.747), and O3 (−0.905).

#### 3.9.4. [Cd(S)(Q)(H_2_O)]

Figure 9C shows the optimized structures of the metal complex [Cd(S)(Q)(H_2_O)] as the lowest-energy configurations. The Cd atom is six-coordinate in octahedral geometry with N1, N2, Cl, and O2, being almost coplanar, deviating by −7.655° (Appendix A).

The N1**•••**N2, N1**•••**O2, and N2**•••**O2 distances in the free ligand (3.788, 4.261, and 7.003 Å) are longer than those in the metal complex (2.886, 2.874, and 3.349 Å) due to metal complex formation through N1, N2, and O2.

The natural charges on the coordinated atoms derived by NBO analysis are Cd (+1.474), O2 (−0.843), N1 (−0.672), N2 (−0.750), N3 (−0.616), O4 (−0.841), and O3 (−0.987).

#### 3.9.5. [Al(S)(Q)(H_2_O)]^+^

Figure 9D shows the optimized structures of the metal complex [Al(S)(Q)(H_2_O)]^+^ as the lowest-energy configurations. The Al atom is six-coordinate in octahedral geometry, with N1, N2, N3, and O4 being almost coplanar, deviating by −5.790° (Appendix A).

The N1**•••**N2, N1**•••**O2, and N2**•••**O2 distances in the free ligand (3.788, 4.261, and 7.003 Å) are longer than those in the metal complex (2.674, 2.673, and 2.872 Å) due to metal complex formation through N1, N2, and O2.

The natural charges on the coordinated atoms according to NBO analysis are Al (+2.116), O2 (−0.916), N1 (−0.763), N2 (−0.858), N3 (−0.695), O4 (−0.920), and O3 (−1.006).

#### 3.9.6. [La(S)(Q)Cl]H_2_O

Figure 9E shows the optimized structures of the metal complex [La(S)(Q)Cl]H_2_O as the lowest-energy configurations. The La atom is six-coordinate in octahedral geometry with N1, N2, N3, and O3 being almost coplanar, deviating by −5.859° (Appendix A).

The N1**•••**N2, N1**•••**O2, and N2**•••**O2 distances in the free ligand (3.788, 4.261, and 7.003 Å) are longer than those in the metal complex (2.991, 3.022, and 3.674 Å) due to metal complex formation through N1, N2, and O2.

The natural charges on the coordinated atoms derived by NBO analysis are La (+1.247), O2 (−0.572), N1 (−0.277), N2 (−0.567), N3 (−0.289), O3 (−0.572), and Cl (−0.460).

Appendix A shows the molecular electrostatic potential (MEP) surface, identifying the positively (blue) and negatively (red, loosely bound, or excess electrons) charged electrostatic potentials in the molecule. The computed total energy, the highest occupied molecular orbital (HOMO) energies, the lowest unoccupied molecular orbital (LUMO) energies, and the dipole moments for the ligands and metal complexes were calculated and are displayed in Table 7 and Appendix A. The more negative values of total energy for the metal complexes than those of the free ligands indicate that the metal complexes are more stable than the free ligands. Furthermore, the energy gaps (Eg = ELUMO − EHOMO) were found to be smaller in the metal complexes than that of the ligand due to chelation of the ligand to the metal ions. The lower Eg in the metal complexes compared to that of the ligand explains the charge–transfer interactions upon metal complex formation.

### 3.10. Antimicrobial Bioassay

A much greater number of drugs are active against Gram-positive than Gram-negative bacteria [47]. In this study, the titled compounds are active against both types of bacteria, which may indicate broad-spectrum antibacterial properties.

The synthesized salen ligand and its mixed 8-hydroxyquinoline metal complexes of Co(II), Ni(II), Cd(II), Al(III), and La(III) were screened for their antibacterial activities against *E. coli* and *P. virgules* (Gram-negative bacteria), and *S. aureus* and *B. subtilis* (Gram-positive bacteria), as well as for antifungal activities against *C*. *albicans* and *A. favas*. The agar diffusion technique was used to evaluate the antibacterial activities of the investigated compounds. The biological activities of the salen ligand, metal salt: (CoCl_2_6H_2_O), (NiCl_2_6H_2_O), (CdCl_2_H_2_O), (AlCl_3_6H_2_O), and (LaCl_3_7H_2_O), and its mixed 8-hydroxyquinoline metal complexes were compared with those of the standard drugs gentamicin as an antibiotic and ketoconazole as antifungal agent.

The results are listed in Appendix A and Figure 10A–C. These data show that mixed salen metal complexes were more potent in inhibiting the growth of microorganisms than the salen ligand. Cd(II)-salen/8-hydroxyquinoline metal complex had a stronger antibacterial efficacy against all bacterial strains except *Escherichia coli* (−ve); its inhibition zone was larger than that of the reference drug gentamicin. The inhibition potencies of mixed ligand metal complexes follow the order: AlSQ > LaSQ > CdSQ > gentamycin > NiSQ > CoSQ > Q > S, LaSQ > CdSQ > gentamycin > AlSQ > NiSQ > CoSQ > Q > S, gentamycin > CdSQ > AlSQ > LaSQ > NiSQ > CoSQ > Q > S and CdSQ > LaSQ > gentamycin > AlSQ > NiSQ > Q > S > CoSQ against *P. vulgaris* (−ve), *E. coli* (−ve), *B. subtilis* (+ve), and *S. aureus* (+ve). The antifungal activity of the mixed salen/8-hydroxyquinoline metal complexes were tested against *C. albicans* and *A. flavus*. The obtained results revealed that all the compounds inhibited the growth of the studied fungi except CoSQ and NiSQ, which had no antifungal activity. Differences in cell wall structure can lead to differences in antibacterial susceptibility, with some antibiotics able to kill only Gram-positive pathogens [47]. For example, Gram-positive bacteria possess a thick cell wall containing many layers of peptidoglycan and teichoic acids, while Gram-negative bacteria have relatively thin cell walls consisting of a few layers of peptidoglycan surrounded by a second lipid membrane containing lipopolysaccharides and lipoproteins.

The variation in the activities of metal complexes against different organisms depends on the impermeability of the microorganism cells or on differences in their ribosomes [48]. Furthermore, the size of the inhibition zone depends upon the concentration of the antibacterial agent where other factors such as culture medium, incubation conditions, and rate of diffusion have been fixed. The activities of all the tested metal complexes may be explained in terms of chelation theory because the free ligand and metal ion have less antimicrobial activity, while at their connection to form metal complex the activity increased [49,50]; moreover, chelation reduces the polarity of the metal atom, mainly because of partial sharing of its positive charge with the donor groups and possible p-electron delocalization over the chelate ring. Furthermore, chelation increases the lipophilic nature of the central atom, which favors its permeation through the lipid layer of cell membranes [51]. The importance of these results lies in the fact that these metal complexes could be applied in the treatment of some common diseases caused by *E. coli*, e.g., septicemia, gastroenteritis, urinary tract infections, and hospital-acquired infections [52]. The noted enhancement in the activity of MSQ compared to their MS binary metal complexes as well as the conceptually and structurally related mixed salen/imidazole metal complexes (MSI) [44] being directly related to introducing 8-hydroxyquinoline (Q) moiety as a secondary ligand. Moreover, probing the antimicrobial activity of the prepared MSQ metal complexes along with their mechanisms of action revealed that the high lipophilicity of Q facilitates the penetration of bacterial cell membranes to reach its target site of action. This most likely to be a metal-binding site of bacterial enzymes. In this respect, the MSQ metal complexes is assumed to undergo a dissociation reaction to liberate a positively charged MS species and free Q ligand [29]. Then, the charged MS species may bind and block the metal-binding sites on bacterial enzymes, thereby inducing the antimicrobial effect [31]. Thus, the lipophilicity is considered to be a crucial factor for antimicrobial activity of the investigated MSQ metal complexes. Moreover, the dissociated free Q ligand has high chelating affinity that enabled its binding to the metallic prosthetic groups of microbial enzymes, thus leading to inhibition of enzymatic activity.

### 3.11. Cytotoxicity

The in vitro cytotoxicity of the mixed metal complexes against human liver carcinoma Hep-G2 cells and breast carcinoma cell line MDA-MB 231 were determined using MTT assay, in which mitochondrial dehydrogenase activity was measured as an indication of cell viability. The absorbance values were analyzed by non-linear regression to obtain IC_50_ values for the five different metal complexes against both cancer cell lines.

The cytotoxicity study of the salen ligand and its mixed 8-hydroxyquinoline metal complexes of Co(II), Ni(II), Cd(II), Al(III), and La(III) against HepG-2 and MDA-MB 231 at concentrations of 0, 0.1, 1, 10, and 100 µM are shown in (Appendix A). On the basis of the results of the surviving fraction of the different compounds and their IC_50_ values, we have displayed the activity of salen and its metal complexes in Table 8.

Figure 11A shows the IC_50_ values of salen and its mixed metal complexes with 8-hydroxyquinoline against the HepG-2 cancer cell line. From these data, we can conclude that salen (IC_50_ = 1.1 µM) and its CoSQ metal complex (IC_50_ = 1.49 µM) have the greatest activities against the HepG-2 cancer cell line. The IC_50_ values of salen and its Co metal complex are lower than that of the standard drug cisplatin (1.55 µM). The IC_50_ values of the metal complexes CdSQ, AlSQ, and LaSQ were found to be 6.19, 5.72, and 1.95 µM, respectively, which were of the same order of magnitude as the reference drug cisplatin (1.55 µM).

Figure 11B shows the cytotoxicity of salen and its mixed 8-hydroxyquinoline metal complexes against the breast cancer carcinoma cell line MDA-MB 231. The IC_50_ values of the metal complexes CdSQ (1.95 μM), AlSQ (2.66 µM), and LaSQ (1.43 μM) were lower than that of salen (27.54 µM). This means that the Cd(II) metal complexes [53,54,55] exhibited the highest activity against breast cancer cell line MDA-MB231. The IC_50_ values followed the order LaSQ < cisplatin < CdSQ < AlSQ < salen.

From the obtained results, we are able to see that the new mixed ligand compounds are potent drugs against human liver carcinoma cell line HepG-2 and breast cancer carcinoma cell line MDA-MB 231, especially the metal complexes CoSQ and LaSQ, which are more potent than the reference drug cisplatin and show concentration-dependent effects, indicating the potential of these metal complexes in cancer therapy. More importantly, the prepared metal complexes are highly specific towards cancer cell lines only, as they exhibited mild toxicity against normal cell line (HEK-293), as illustrated in Table 8 and Appendix A.

### 3.12. Molecular Docking Studies

Molecular docking studies were performed using MOA2014 software [32] in order to explore the possible binding modes for the most active site of the receptor of breast cancer oxidoreductase (PDB ID: 3HB5). The crystal structures of the 3HB5 breast cancer receptor were downloaded from the Protein Data Bank (http://www.rcsb.org./pdb accessed date 21 April 2021). The structures of salen and its metal complexes were created in PDB file format using the Gaussian09 software package.

The binding affinities of the ligand and the synthesized CdSQ metal complexes against the receptor of 3HB5 are shown in Figure 12 and Figure 13 and Table 9. The binding energies of the ligand and its metal complexes with the target protein receptor follow the order CdSQ > salen, which agrees with the experimental results.

## 4. Conclusions

In the current work, we studied the anti-cancer and anti-microbial effect of Co(II), Ni(II), Cd(II), Al(III), and La(III) metal complexes of mixed ligand(2,2′-{1,2-ethanediylbis[nitrilo(E) methylylidene]}diphenol with 8-hydroxy quinolone. The quantum calculation, analytical, and experimental measurements seem to suggest the proposed structure of the compounds. Molecular docking studies were performed to investigate the binding of the synthesized compounds with breast cancer oxidoreductase (PDB ID: 3HB5). The outcomes can be abbreviated as follows:The metal complexes exhibited high cytotoxic potency against human breast cancer (MDA-MB231) and liver cancer (Hep-G2) cell lines.The anticancer results showed that the IC_50_ values of CoSQ and LaSQ were 1.49 and 1.95 µM, respectively, which is comparable to that of cisplatin (1.55 µM) against Hep-G2 cells. On the other hand, CdSQ and LaSQ were the most effective against MDA-MB231, with IC_50_ values of 1.95 and 1.43 µM, respectively.The antibacterial and antifungal efficacies for the MSQ metal complexes decreased in the order AlSQ > LaSQ > CdSQ > gentamycin > NiSQ > CoSQ > Q > S for antibacterial activity, and for antifungal activity followed the trend LaSQ > AlSQ > CdSQ > ketoconazole > NiSQ > CoSQ > Q > S.The results of conductivity confirm that the solutions were nonelectrolytes, except that of the Al metal complex, which was found to be ionic.The average particle sizes for mixed ligand metal complexes with Cd(II) and La(III) were calculated to be 0.423 and 0.343 nm, respectively.On the basis of DFT calculations, we found the geometry of prepared metal complexes to be octahedral.

## Data Availability

The data presented in this study are available on request from the corresponding author.

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
