# Peer review of "Synthesis, DFT Calculations, Antiproliferative, Bactericidal Activity and Molecular Docking of Novel Mixed-Ligand Salen/8-Hydroxyquinoline Metal Complexes"

_molecules, 2021, doi:10.3390/molecules26164725_

Round 1
Reviewer 1 Report
This is a resubmitted version of a manuscript that I have been reviewing before. Although my previous report was very detailed, the Authors have managed to improve and correct their manuscript so that the level of the current version can be accepted.
However, there is still one thing that needs to be revised and is related to the DFT calculations. The Authors have not stated if they have calculated the IR frequencies in order to find out if the structures are at the true minima. I hope that they did but just forgot to state this in the Materials&Methods section. This is particularly important since the difference between the energies of tautomeric forms are significant.
Author Response
Response: IR Frequency calculations were performed on the optimized geometries, show that all the converged states correspond to true minima (no imaginary frequencies). This was added to experimental part.
Reviewer 2 Report
This contribution of L. H. Abdel Rahman reports the synthesis, characterization, modeling and study of properties of complexes of Co(II), Ni(II), Cd(II), La(III) and Al(III) with deprotonated salen and 8-hydroxyquinoline.
Despite the variety of characterization techniques employed, I found that structures shown in Scheme 1 for the complexes are not based on experimental facts. First, the bonding mode proposed for the ligand as tridentate with a non-aromatic ring is really uncommon for the salen ligand, and the authors have to explain why in this particular case the ligand adopts this bonding mode instead of the more conventional tetradentate N2O2. Just to mention, there are almost 3400 X-ray structures in the CCDC in which salen bonds as tetradentate N2O2, but none containing the ligand as proposed by the authors. In Scifinder this bonding mode is neither reported, although Schiff Bases from cyclohexadienone are known and commercial. There is probably a good reason at the level of the complex to explain this anomaly in the bonding mode, and the authors should explain this rare deviation. In the same respect, if there are other examples (others beyond those of the same authors), they should be referenced. However, the rarity of this bonding is not the problem, the problem is the experimental support.
The experimental data provided to support these structures are not consistent with the structures themselves. The NMR spectra of 1H (not provided in all cases, not even in the supporting) should show at least 10 different signals corresponding to the aromatic protons of the salen (all of them are chemically non equivalent), and another 6 from the hydroxyquinoline fragment (also non equivalent). Nothing of that is observed, there are much less signals indicating (probably) a higher symmetry. The same is observed for the central CH2CH2 group. DMSO is not probably the best solvent for this tpe of complexes. Did the authors try CDCl3? (commercial Co(salen)2 is soluble in chloroform). Maybe there is a reason for this equivalence (fluxionality, labile behaviour) but nothing of that has been explained. The same happens with the 13C spectra, where the "spectra" show really much less signals than expected: all carbon nuclei are non equivalent, so 25 peaks are expected. However spectrum B shows 10 different peaks, while spectrum C shows 13-14 peaks (it is not clear...). So the 13C NMR neither reflects the proposed structure in no case. Particularly relevant is the region of 160-200 ppm, where if a non-bonding cyclohexadienone is proposed, a clear peak at about 180-185 ppm should be observed, and spectra B and C don't show that. If concentration is a problem, NMR offers solutions (HMQC or HMBC experiments) to detect these insensible nuclei several bonds apart. Moreover, there is an additional (serious) problem here: B and C spectra look like as "normal" spectra, while D and E clearly do not correspond to "normal" spectra, because this shape is not usual in 13C spectra. How were they acquired? Why do they look so different?
In summary, a quite strange bonding mode has been proposed for the salen ligand, whose formation in the reaction conditions is not justified and, moreover, its presence is not supported by the experimental data (NMR). I'm sure that the compounds contain one metal, one salen, and one hydroxyquinoline, but not in this bonding mode, which has to be proved. There are additional inconsistencies, such as (i) why a water molecule is bonded to the metal, but there is a free oxygen? the chelate effect is not acting anymore?; (ii) the upper structure in Scheme 1 is valid for all Co(II), Ni(II), Cd(II) AND Al(III)? Are not there differences in the case of Al(III), considering the different charge?; (iii) the La(NO3)3 in the lower part should be replaced by LaCl3, otherwise, where the bonded chloride come from?
The correct structural characterization of the prepared compounds is one of the key elements of the research. In absence of a correct structural determination, further conclusions are not valid. In this case DFT analysis, modeling, docking (and so on) are not valid, because the starting point is not proved. Based on these grounds, this contribution has to be rejected.
Author Response
Reviewer 2
- First, the bonding mode proposed for the ligand as tridentate with a non-aromatic ring is really uncommon for the salen ligand, and the authors have to explain why in this particular case the ligand adopts this bonding mode instead of the more conventional tetradentate N2O2. Just to mention, there are almost 3400 X-ray structures in the CCDC in which salen bonds as tetradentate N2O2, but none containing the ligand as proposed by the authors. In Sc finder this bonding mode is neither reported, although Schiff Bases from cyclohexadienone are known and commercial. There is probably a good reason at the level of the complex to explain this anomaly in the bonding mode, and the authors should explain this rare deviation. In the same respect, if there are other examples (others beyond those of the same authors), they should be referenced. However, the rarity of this bonding is not the problem, the problem is the experimental support.
Response:
All got data strongly confirm keto-enol form rather than the usual enol-enol form. Although this finding is unexpected, something can obviously note it in the IR analysis of all complexes, where the detection of carbonyl group around 1740 cm-1 and the disappearance of OH group signature is evident. Similarly, 1HNMR confirmed the presence of NH group at ~13 ppm. These experimental findings, along with the non-electrolytic nature of the complexes (owing to Cl-coordination) lend powerful support to the suggested structures of the complexes. Indeed, X-ray crystal structure would be ideal and direct solid evidence for the proposed structure; however, all our efforts to grow single crystals for all prepared complexes were unsuccessful. However, DFT calculations could provide clear-cut evidence about the possible favored structure of prepared complexes. In doing so, the total energies calculations have been performed taking into accounts that salen ligand could possibly have either enol-enol or enol-keto form, where the latter was stabilized by strong resonance-assisted hydrogen bonding induced by the coordination of chloride ion to the metal center in the inner sphere of the complex. This result supports the experimental findings in which the salen ligand acts as a mono basic tridentate.
We find similar coordination mode in the literature:
B.I. KHARISOV, M.A. ME´ NDEZ-ROJAS, A.D. GARNOVSKII, E.P. IVAKHNENKO and U. ORTIZ-ME´ NDEZ, J. Coord. Chem., 2002, Vol. 55(7), pp. 745–770 REVIEW
The experimental data provided to support these structures are not consistent with the structures themselves. The NMR spectra of 1H (not provided in all cases, not even in the supporting) should show at least 10 different signals corresponding to the aromatic protons of the salen (all of them are chemically non aromatic protons of the salen (all of them are chemically nonequivalent), and another 6 from the hydroxyquinoline fragment (also non equivalent). Nothing of that is observed, there are much less signals indicating (probably) a higher symmetry. The same is observed for the central CH2CH2 group. DMSO is not probably the best solvent for this type of complexes. Did the authors try CDCl3? (commercial Co(salen)2 is soluble in chloroform). Maybe there is a reason for this equivalence (fluxionality, labile behaviour) but nothing of that has been explained. The same happens with the 13C spectra, where the "spectra" show much less signals than expected: all carbon nuclei are non-equivalent, so 25 peaks are expected. However spectrum B shows 10 different peaks, while spectrum C shows 13-14 peaks (it is not clear...). So the 13C NMR neither reflects the proposed structure in no case. Particularly relevant is the region of 160-200 ppm, where if a non-bonding cyclohexadienone is proposed, a clear peak at about 180-185 ppm should be observed, and spectra B and C don't show that. If concentration is a problem, NMR offers solutions (HMQC or HMBC experiments) to detect these insensible nuclei several bonds apart. Moreover, there is an additional (serious) problem here: B and C spectra look like as "normal" spectra, while D and E clearly do not correspond to "normal" spectra, because this shape is not usual in 13C spectra. How were they acquired? Why do they look so different?
Response:
First, thanks for your positive comment. We have tried to use CDCl3 solvent; unfortunately, none of these compounds have been dissolved. So, we had to use DMSO solvent as it was the only one suitable solvent for all these complex compounds. As for disappearance of some important peaks in 13CNMR; this is owing to long time needed for analysis besides weak solubility of these compounds. That led to disappearance of some important peaks in 13CNMR only but other analysis methods like 1HNMR, IR spectra, elemental analysis and Mass analysis confirmed our suggested structures.
In summary, a quite strange bonding mode has been proposed for the salen ligand, whose formation in the reaction conditions is not justified and, moreover, its presence is not supported by the experimental data (NMR). I'm sure that the compounds contain one metal, one salen, and one hydroxyquinoline, but not in this bonding mode, which has to be proved. There are additional inconsistencies, such as (i) why a water molecule is bonded to the metal, but there is free oxygen? the chelate effect is not acting anymore?; (ii) the upper structure in Scheme 1 is valid for all Co(II), Ni(II), Cd(II) AND Al(III)? Are not there differences in the case of Al(III), considering the different charge?; (iii) the La(NO3)3 in the lower part should be replaced by LaCl3, otherwise, where the bonded chloride come from?
Response:
The carbonyl group did not take a part in the coordination process as it is concluded from the IR spectral results of the salen ligand and the complexes, where the stretching vibration of the carbonyl group Ê‹ (C=O) appeared in the same place for all complexes at 1740 cm-1. Also, the DFT calculations revealed that the phenyl ring containing the carbonyl group does not lie in the same plane of the of the salen molecule due to the free rotation. The IR spectra of the Co(II), Ni(II), Cd(II), and Al(III) metal complexes show bands at 950, 903, 976, and 980 cm -1, respectively, which are assigned to the rocking mode of coordinated water (Table 3).
Scheme 1 has been revised and Al(III) complex has been drawn with its Cl- counter ion to neutralize the charge.
All the analytical tools confirm this structure, and this work is an extension of a previous work. ref [37], https://doi.org/10.1002/aoc.5912
Reviewer 3 Report
Please see attached pdf.

Author Response
Reviewer 3
General comment:
- The authors claim that “The complexes exhibited high cytotoxic potency against human breast cancer (MDA-MB231) and liver cancer (Hep-G2) cell lines” It is incredible that Cadmium and other heavy metal complexes have anticancer properties when these metals are cancer-causing themselves. Specifically Cd is carcinogenic. The authors assume that because the ligands have been found to have anticancer properties, their metal complexes may also have. Is there any such example in the literature? If yes it must be cited.
Response: I agree with you that the free cadmium ions are toxic. Upon coordination with some organic donor ligands to form complexes, it is found that it has anti-cancer properties. The obtained results in this work of this is supported by the results obtained in the Ref. [53-55].
Specific comments:
- Throughout the text metal complexes are referred as “complexes”. This is incorrect substitute with “metal complexes”.
Response: All the complexes has been replaced by metal complexes throughout the text.
Complexes of the a forementioned metals with the two ligands have been synthesized before. There are no anticancer properties found. What makes the authors believe that if they mix the ligands, the new complexes will have anti-cancer properties?
Response:
Mixed ligand complexes of Schiff base which have nitrogen, oxygen and sulfur donor atoms in their backbones play an important role in biological process as exemplified by many instances in which enzymes were known to be activated by metal ions. Among mixed ligand complexes derived from quinoline and quinoline derivative quinolinones are the most important class of heterocyclic compounds and a part of the alkaloid family [1]. Quinolines have emerged as potential therapeutic agents because of their conformational rigidity and improved physical properties, such as charge density or lipophilicity, and pharmacological advantages such as metabolic stability and oral bioavailability [1, 2]. Quinolines, considered as the backbone for most natural products, were used for the design of many synthetic compounds having diverse pharmacological applications such as antibacterial, antifungal [3. 4], anticancer [5], antimalarial and antiviral [6].
1.Marcaccino S, Pepino R, Pozo MC, Basurto S, Garia-valverde M, Torroba T. One-pot synthesis of quinolin-2-(1H)-ones via tandem Ugi-Knoevenagel condensations. Tetrahedron Lett. 2004;45:3999–4001.
- Kulkarni NV, Hegde GS, Kurdekar GS, Budagumpi S, Sathisha MP, Revankar VK. Spectroscopy, electrochemistry, and structure of 3d-transition metal complexes of thiosemicarbazones with quinoline core: evaluation of antimicrobial property. Int J Rapid Commun. 2010;43:235–46.
- Sudha N, Selvi G. Synthesis, characterization, and biological studies on Fe(II) and Zn(II) quinoline Schiff Base complexes. Int J Chem Tech Res. 2015;8:367–74.
- Solomon VR, Lee H. Quinoline as a privileged scaffold in cancer drug discovery. Curr Med Chem. 2011;18:1488–508.
- Bentzinger G, De SW, Mullié C, Agnamey P, Dassonville-Klimpt A, Sonnet P. Asymmetric synthesis of new antimalarial aminoquinolines through Sharpless aminohydroxylation. Tetrahedron Asymmetry. 2016;27:1–11.
- Anantacharya R, Manjulatha K, Satyanarayan ND, Santoshkumar S, Kaviraj MY. Antiproliferative, DNA cleavage, and ADMET study of substituted 2-(1-benzofuran-2-yl) quinoline-4-carboxylic acid and its esters. Cogent Chem. 2016;2:2016. doi:10.1080/23312009.1158382.
“Laboratory temperature” = what is the actual temperature so that the person who wants to reproduce the experiment needs to work with?
Response: 79 °C
- The scheme 1 structures are not drawn properly with the arrows not positioned properly (e.f. the arrow starting at NH). Please use ChemDraw and redraw correct the bonds. Also, the letters have all kinds of fonts in the scheme.
Response: Scheme 1 has been revised and redrawn.
- The integrations in the NMR are not precise. Please redo them. Also set the reference peak for integration with a whole number so that we can count the protons in respect to it.
Response: The integration has been done as requested
There are yellow highlights throughout the text that I am not sure what they serve.
Response: The highlights has been removed throughout the text.
- The NMRs need to go to the experimental. There are unaccounted peaks in Figure 1 NMR. Please explain. The same for mass spectrum.
Response: We have added the unaccounted peaks. Please check the yellow highlighted ones in experimental part.
- Indicate the molecular peak in the mass spectrum.
Response: 267.65 in NiSQ and 266.27 in CdSQ
Round 2
Reviewer 3 Report
Thank you for the corrections.
Author Response
Molecules-1283722
“Synthesis, DFT Calculations, antiproliferative, bactericidal activity and molecular docking of novel mixed-ligand Salen/8-hydroxyquinoline metal complexes”
Dear Editor-in-Chief of Journal of Molecules,
Thank you very much for giving us the opportunity to review our manuscript to present it in its best form. Here, we submit a new version of our manuscript, which has been changed according to the requested suggestions. We have marked the changes in yellow highlight color in the new version manuscript
RESPONSE TO EDITORS COMMENTS:
In the absence of an X-ray crystal structure and with ambiguous NMR data, the conclusions about the structure of the complex must be downgraded, with expressions like ... these results seem to suggest that ..., and recognizing that more solid experimental data such as X-ray crystal structures and better NMR data would be needed to more consistently confirm the proposed structure. This should be done throughout the text. Regarding the biological part, the cytotoxicity of the compounds on normal cells should be included in a table in the main text, together with the cytotoxicity to cancer cells lines and the corresponding selectivity indices. Finally, international nonproprietary names such as gentamycin and ketoconazole must be written non-capitalized.
Response: done as requested
This manuscript is a resubmission of an earlier submission. The following is a list of the peer review reports and author responses from that submission.
Round 1
Reviewer 1 Report
The submitted manuscript present the results of synthesis and analysis (both experimental and theoretical) of the series of metal complexes. While the amount of presented information is large, the style of presentation is very poor. Also, there are some major issues about the inappropriate use of the methods which may lead to false results. The manuscript should undergo MAJOR revision to be considered for possible publication.
Continuous line numbering would greatly improve and facilitate the review process. Besides, it is mandatory in molecules. Therefore, I will point only the page which my comment refers to.
Keywords: please unify the font
Page 1, first line, do you mean metaloorganic complexes or complexes in general?
Page 2, “A BRUKER model 400 MHZ”, this is not a model, it is 1H frequency, the model can be i.e. AVANCE. Please clarify.
Table 1, please round molecular weight of salen. Experimental elemental composition should be compared with the theoretical one (based on the supposed formula).
Page 5, “and stable towards air and moisture.” this statement is based on what? Have you performed any experiments to prove this thesis?
Page 5, “are consistent with their proposed molecular formula” where is this comparison presented?
Figure 1, the quality of this figure is very poor, especially the DPI. Please improve it.
Table 2, since you have been doing DFT calculations why you have not calculated NMR shielding constants in order to 1)prove the formation of complexes 2)prove the spectral assignments. This MUST be corrected.
The quality of Figure S1 is very poor. Why the spectra were horizontally expanded? They look really bad.
Scheme 2 should be moved to the SI.
Figure 6, PXRD, please remove the background from the patterns. Also, please register the PXRD of the physical mixture of the reactants and compare it with the PXRD of complexes and pure reactants. This is the only way to prove the complex formation (via PXRD).
Line 15, are you sure that the keto-enol form is stable by more than 350 kcalmol??? This is a very large energy difference. This is probably a mistake.
Table 7, please state the unit of energy, not a.u.!
Figure 9 and 10, please move them to SI.
Page 21, please unify the font and style, this page looks bad.
Page 23, aren’t you afraid about this very high toxicity? And its potential disadvantages? There is no single word about this in discussion.
Page 24, molecular docking must be described in more details. Please describe this method clearly in the experimental section. Especially the forcefield and the types of atoms used for metals.
Conclusions: “Based on DFT calculations the geometry of prepared complexes is the octahedral.”-this is because you have optimized the initially octahedral complexes. What you must do is suggest another structures and optimize them at the same level of theory and then compare the energies, don’t forget about calculating NMR!
Part describing each Author’s individual contribution is missing. This is mandatory in molecules.
Reviewer 2 Report
The idea reported on the presented manuscript, considering the exploitation of the use of mixed salen-8-HQ metal complexes as antiproliferative and bactericidal agents seems very interesting. Nevertheless, the work done and mainly its presentation has some serious drawbacks.
In particular, the present manuscript is very similar to another one of the same authors, "Synthesis, Characterization, Theoretical Studies, and Antimicrobial/Antitumor Potencies of Salen and Salen/Imidazole Complexes of Co (II), Ni (II), Cu (II), Cd (II), Al (III) and La (III).", Appl Organomet Chem.2020;34:e5912.
Moreover, there is relevant information missing (like the theoretical values for the Elemental Analysis, the NMR and spectroscopic data for 8-HQ, etc.). ESI-MS data, in my opinion, is not informative at all; not even the peaks of the free ligands are observed what is, at least, strange.
Then, considering the biological tests, the major problem that I report is that the metal pure salts were never used as control. In this way, is not possible to be sure if the observed effects are relative to the mixed complex or the metal itself. Without this knowledge, the sentence on 4th paragraph of pag 21, starting as "The activities of all the tested complexes may be explained based on chelation theory...." can not be used. Keeping our attention in the same paragraph, uptake and lipophilicity studies are also need to corroborate the discussion written wherein.
Still concerning cytotoxicity studies, IC50 value for LaSQ against MDA-MB231 cell line should be higher than the one for AlSQ and CdSQ complexes (according to Fig S5). This should be revised.
All this are only some fundamental issues that led me to suggest the rejection of the present manuscript for publication.